# An Oxalate Transporter Gene, *AtOT*, Enhances Aluminum Tolerance in *Arabidopsis thaliana* by Regulating Oxalate Efflux

**DOI:** 10.3390/ijms24054516

**Published:** 2023-02-24

**Authors:** Zongming Yang, Pingjuan Zhao, Xuehua Luo, Wentao Peng, Zifan Liu, Guishui Xie, Mengyue Wang, Feng An

**Affiliations:** 1Hainan Danzhou Agro-Ecosystem National Observation and Research Station, Rubber Research Institute of Chinese Academy of Tropical Agricultural Sciences, Danzhou 571737, China; 2College of Tropical Crops, Hainan University, Haikou 570228, China; 3Institute of Tropical Bioscience and Biotechnology, Chinese Academy of Tropical Agricultural Sciences, Haikou 571101, China

**Keywords:** *Arabidopsis thaliana*, AtOT, aluminum toxicity, oxalate transporter, gene expression, functional characterization

## Abstract

Secretion and efflux of oxalic acid from roots is an important aluminum detoxification mechanism for various plants; however, how this process is completed remains unclear. In this study, the candidate oxalate transporter gene *AtOT*, encoding 287 amino acids, was cloned and identified from *Arabidopsis thaliana*. *AtOT* was upregulated in response to aluminum stress at the transcriptional level, which was closely related to aluminum treatment concentration and time. The root growth of *Arabidopsis* was inhibited after knocking out *AtOT*, and this effect was amplified by aluminum stress. Yeast cells expressing *AtOT* enhanced oxalic acid resistance and aluminum tolerance, which was closely correlated with the secretion of oxalic acid by membrane vesicle transport. Collectively, these results underline an external exclusion mechanism of oxalate involving *AtOT* to enhance oxalic acid resistance and aluminum tolerance.

## 1. Introduction

In acidic soil (pH < 5), active AlOH^2+^ and Al^3+^ ions will be adsorbed on the cation exchange sites on the surface of soil particles or dissolved into the soil solution, thereby causing aluminum toxicity and harmful effects on plant growth and soil microbial activities [1,2,3]. With the deterioration of the soil environment and the application of chemical fertilizers, the degree of soil acidification is further deepened and the gross area of soil acidification is further expanded, thereby making aluminum toxicity a non-negligible factor in crop production [4]. Therefore, it is vital to study the mechanism of aluminum toxicity and detoxification in plants, which is one of the hotspots of plant stress biology [5,6].

External exclusion and internal tolerance mechanisms have been extensively studied as the main aluminum detoxification mechanism in plants [7,8]. Organic acids play an important role whether in the external exclusion mechanism or internal tolerance mechanism [9,10,11,12,13,14,15,16]. Citric acid, malic acid, and oxalic acid are the three main organic acids related to aluminum detoxification [17,18,19,20]. Thus far, malic acid and citric acid transporter genes have been found in various plants, including *AtMATE* [21] and *AtALMT1* [22] in *Arabidopsis*, *OsFRDL2* [23] and *OsFRDL4* [24] in rice, *BnALMT1* and *BnALMT2* [25] in rape, *ScALMT1* [26] in the rye, *GmALMT1* [27] in soybean, *SbMATE* [28] in sorghum, *TaALMT1* [29] in wheat, and so on. Nevertheless, although the secretion of oxalic acid from roots has been reported as an important mechanism of aluminum detoxification in taro, buckwheat, tea, spinach, tomato, polygonum, and so on [14,15,17], excepting that Lv et al. (2021) screened an aluminum-induced expression of the oxalate secretion-related regulatory gene *MsDHN1* through transcriptome analysis of alfalfa under aluminum stress; the oxalate transporter in plants has not been identified and reported [30].

*Arabidopsis*, as the model plant, has the advantages of a short growth cycle, complete genome information, and a high genetic transformation rate [31], which is widely applied in the discovery and identification of plant aluminum tolerance genes [32,33,34,35]. Several studies have highlighted the role of the aluminum-activated malate transporter (ALMT), the multidrug and toxic compound extrusion (MATE), and some transcription factors (STOP1 [36], WRKY [37], NADP-ME [38], etc.) in regulating the exclusion of citric acid and malic acid in aluminum detoxification. Meanwhile, oxalic acid is generally reported as a factor hindering plant growth in *Arabidopsis* [39,40], which is one-sided.

In our previous study of aluminum detoxification in rubber tree (*Hevea brasiliensis*), two oxalate transporter candidate genes *HbOT1* and *HbOT2* were identified by yeast AD12345678 referencing the study of the oxalate transporter in brown rot fungus *Fomitopsis palustris* [41,42]. Meanwhile, a new gene highly homologous to *F. palustris* oxalate transporter gene *FpOAR* and *HbOT1* and *HbOT2* was screened out in *Arabidopsis* and named *AtOT* (*A. thaliana* oxalate transporter). Herein, for the bio-informatics and expression analyzes, a yeast AD12345678 system and an *Arabidopsis* mutant were used to identify whether or not *AtOT* performs an oxalic acid transport function and whether or not its function is related to aluminum detoxification.

## 2. Results

### 2.1. AtOT Cloning and Sequence Analysis

An unknown protein NP_192696.3 of *Arabidopsis* and the corresponding candidate gene NM_117026.5 were obtained based on the identified oxalate transporter FpOAR (GeneBank: BAJ10704.1) of *F. palustris* and temporarily named *AtOT* (ID_NCBI: AT4G09580). The full length of *AtOT* is 1374 bp, with the CDS region size of 864 bp and encoding 287 amino acid residues. The ORF sequences of *AtOT* were cloned from the Col-0 cDNA library by PCR (Appendix A).

The basic physicochemical properties of *AtOT* protein were predicted by ProtParam, with a molecular weight (MW) of 28,630.58 Da, an isoelectric point (PI) of 9.61, an instability index (II) of 33.70, and a total average hydrophilicity (GRAVY) of 0.473. There were five transmembrane helixes at 53–75, 111–133, 143–165, 226–248, and 263–280 of the *AtOT* amino acid sequence. In addition, the hydrophilic/hydrophobic analysis showed that the hydrophobic parts of *AtOT* were greater than the hydrophilic parts, which indicates that *AtOT* has the basic properties of a transporter protein. Meanwhile, as predicted by PSORT, *AtOT* is likely to be a membrane-localized protein, which provides a prerequisite for *AtOT* to perform the transmembrane transport function of oxalate. In addition, the *AtOT* protein had a SNARE-assoc domain, located in the 104–284 N-terminal, with an E-value of 4.86 × 10^−16^ by NCBI CDD database and SMART (Figure 1).

Through the NCBI BLASTP search, *AtOT* had higher homology with SNARE proteins from *Capsella rubella* (XP_006288372.1), *Arabidopsis lyrata* subsp. *Lyrata* (XP_002874575.1), *Arabis alpina Linn.* (KFK32071.1), *Eutrema salsugineum* (XP_006397182.1, XP_024010027.1), *Brassica carinata* (KAG2330225.1), *Sinapis alba* (KAF8106695.1), *Brassica rapa* (RID45809.1, XP_009113534.1), *Brassica napus* (XP_013656110.2, XP_013731370.2, KAH0858802.1, KAH0910707.1), and *Hevea brasiliensis* (XP_021655511.1). Among them, *AtOT* had the highest homology with *Arabidopsis lyrata* subsp. *Lyrata* SNARE protein XP_002874575.1 and *Arabis alpina Linn* SNARE protein KFK32071.1, which were 96.26% and 94.12%, respectively (Appendix A). However, all of these SNARE proteins had not been identified, which indicates that further experiments must be carried out to prove the possible function of *AtOT*.

### 2.2. Expression Patterns of AtOT

Quantitative real-time PCR (qRT-PCR) was used to determine the transcript profile of *AtOT*. For Al^3+^ stress, the *Arabidopsis* seedlings with two true leaves were treated with AlCl_3_·6H_2_O on 1/2 MS solid medium (Figure 2A). As shown in Figure 2B, *AtOT* presented a constitutive expression pattern, which was richest in the root, followed by the stem, and lowest in the leaf. Following treatment with different concentrations of Al^3+^, expression levels of *AtOT* showed a significant upregulation from 50 μM Al^3+^ to 200 μM Al^3+^ and peaked at 100 μM Al^3+^ with an approximately 4.51-fold increase in expression (Figure 2C). In addition, *AtOT* expression gradually increased under the time-course treatments of 100 μM Al^3+^ with prolonged treatment and reached its highest value at 48 h (Figure 2D).

### 2.3. Subcellular IzationLocalization of AtOT

The 35S::AtOT-GFP (green fluorescent protein) recombinant was transformed into the tobacco leaf epidermal cells for the subcellular localization of the *AtOT* protein. Observation by an LSM800 confocal laser scanning microscope revealed that 35S::AtOT-GFP was restricted to the plasma membrane only (Figure 3) thereby indicating that *AtOT* is localized in the plasma membrane in contrast to 35S::1300-GFP alone, which exhibited fluorescence throughout the cell.

### 2.4. Reverse Genetic Analysis of AtOT in Arabidopsis

The phenotype, root length, proline content, and MDA content of wild-type Col-0 and an *atot* mutant under different concentrations of Al^3+^ stress were compared to examine whether or not Al-responsive *AtOT* is involved in aluminum detoxification. The homozygosity of the *atot* mutant was firstly confirmed. As shown in Figure 4A, the five *atot* mutant plants showed clear band for LBb1.3 + RP primers but no clear band for LP + RP primers in the PCR; therefore, they were all homozygotes. However, since plant no. 2 and no. 3 showed weak bands for LP + RP primers, to make the experiments more convincing, only the seeds produced from plants no. 1, 4, and 5 were collected as *atot* mutant homozygotes and used for the subsequent experiments. The root lengths of the wild-type and *atot* mutant were 4.03 and 2.74 cm, respectively, under non-stressed conditions, which reflected that knocking out *AtOT* might negatively affect root growth. With the increase in Al^3+^ concentration, the root length of the wild type and *atot* mutant showed a trend of shortening and finally reached the lowest values at 200 μM Al^3+^, which were 1.32 and 0.70 cm, respectively (Figure 4B,C), indicating that aluminum stress was more harmful to the *atot* mutant than to the wild-type. We also measured the contents of proline (Pro) and malondialdehyde (MDA) in the wild type and *atot* mutant under aluminum stress to verify this conclusion. The contents of Pro and MDA in the wild type and *atot* mutant gradually increased with the increase in Al^3+^ concentration, and these contents in the *atot* mutant were significantly higher than those in the wild type under the same concentration of Al^3+^, which proved the positive role of *AtOT* in preventing aluminum-toxicity-induced oxidative damage (Figure 4D,E).

### 2.5. Functional Characterization of AtOT in Yeast

*AtOT* was expressed in the yeast mutant AD1-8 to investigate the function of AtOT in oxalic acid transportation. Moreover, the construct pDR196-*FpOAR* and empty vector pDR196 were used as positive and negative controls, respectively (Figure 5A). Based on the colony phenotype, the yeast strain AD1-8 can grow normally if the concentration of oxalic acid is not higher than 2 mM, which depends on the autologous oxalic acid resistance. Meanwhile, on plates containing 4–8 mM oxalic acid, the *AtOT*-transformed yeast showed clear cell growth which was similar to the positive control, whereas very little growth was observed in the negative control. When the concentration of oxalic acid reached 10 mM, the *AtOT*-transformed yeast could still grow, whereas it was hard for the negative and positive controls to form any colonies. On the other hand, the qRT-PCR showed that *AtOT* expression could be induced by oxalic acid stress and it peaked at 2 mM oxalic acid in yeast cells (Figure 5B). These results implied that *AtOT* enhanced yeast cell oxalic acid resistance through some pathways responding to oxalic acid.

Further yeast transformants were cultured in SD (-Ura) liquid medium containing 2 mM oxalic acid for 13 days. Although the pH value of each transformant decreased with time, the pH value and trend of the positive control and the *AtOT*-transformed yeast were the same and the pH value of the negative control was always higher than that of the positive control and the *AtOT*-transformed yeast under the same culture days (Table 1). At the end of the culture, the pH of each transformant reached the lowest value, with the average pH for the positive control, the *AtOT*-transformed yeast, and the negative control reaching 2.42, 2.38, and 2.76, respectively. Meanwhile, the dry weight of each transformant increased with time, and the overall dry weight was as follows: *AtOT*-transformed yeast > positive control > negative control (Table 2). *AtOT* may affect the oxalic acid metabolism of yeast cells and reduce the inhibition of oxalic acid to yeast cells; hence, the acidity of the bacterial solution significantly decreased and the biomass accumulation increased under oxalic acid stress.

The oxalic acid content in yeast cells was determined by the sulfosalicylic acid method. Oxalic acid stress caused the oxalic acid content in yeast to significantly increase in a short time (≤1 day), thereby indicating that the short-term harmful effect of oxalic acid stress on yeast cells was inevitable. During the 1st to 13th day of culture, the oxalic acid content of the positive control and the *AtOT*-transformed yeast showed a decreasing trend over time (Figure 6A). Finally, the oxalic acid content of the positive control and *AtOT*-transformed yeast was 4.12 mM and 5.83 mM, respectively. However, the oxalic acid content of the negative control was always at a high level. At the end of the culture, it was significantly higher than that of the positive control and *AtOT*-transformed yeast, with it being 11.20 mM.

In addition, the oxalic acid content in the culture medium was determined using the same method as described above. As shown in Figure 6B, the oxalic acid content in the medium of the negative control changed irregularly during the culture time, which showed that oxalic acid stress had caused disorder of oxalic acid metabolism in the negative control. Meanwhile, with the passage of culture time, the oxalic acid content in the medium of the positive control and *AtOT*-transformed yeast generally showed an upward trend and two common points between them that deserve to be noticed. The first common point was that the oxalic acid content in the medium rapidly increased at a certain time point, and the second common point was that the oxalic acid content in the medium was relatively stable for a period after the rapid increase, which further indicated that *AtOT* may be functionally related to oxalic acid efflux.

The [^13^C]oxalic acid residue in the filtrate of yeast membrane vesicles was recorded in the form of δ ^13^C by in an vitro oxalate transport study, which indirectly reflected the absorption of oxalic acid by vesicles. Apparent differences were observed between membrane vesicles of *AtOT*-transformed yeast and the negative control whether MgATP was added or not (Figure 7). The δ ^13^C in the filtrate of the negative control was 5.72 times more than that in the filtrate of the *AtOT*-transformed yeast membrane vesicles when lacking MgATP. In the presence of MgATP, the magnification was increased to 6.01, thereby suggesting that *AtOT* significantly regulated the absorption of oxalic acid by membrane vesicles.

The involvement of *AtOT* in aluminum tolerance was identified by the same method as the oxalic acid stress study. The empty vector pDR196 was used as the negative control in this study as well. At an Al^3+^ concentration below 2.7 mM, the growth of the negative control and the *FpOAR*- and *AtOT*-transformed yeast did not significantly differ. Nevertheless, it was hard for the negative control to form a colony once the Al^3+^ concentration reached 2.7 mM; on the contrary, the *AtOT*-transformed yeast could still grow (Figure 8A). When examining the transcripts of *AtOT*, the amount of *AtOT* transcripts increased 1.58-, 2.26-, 3.15-, and 3.18-fold as compared to that of the control under 2.4, 2.6, 2.7, and 2.8 mM Al^3+^ stresses, respectively (Figure 8B). These results revealed that *AtOT* improved the aluminum tolerance of yeast cells under high concentrations of aluminum.

The growth of each transformant was consistent with the characteristics of the S-shaped growth curve (Figure 8C). The *FpOAR*-transformed yeast and the negative control showed a similar pattern in the growth trend, and their OD_600_ values were finally stable at about 2.10 and 2.24, respectively. Nevertheless, the *AtOT-*transformed yeast entered the logarithmic growth phase at 8 h and reached the platform stage at 18–20 h, with an OD_600_ value of approximately 1.83 at the end of the culture. Although the OD_600_ value of *AtOT*-transformed yeast was slightly lower than that of the negative control and *FpOAR*-transformed yeast at the platform stage, the time that it entered the logarithmic growth phase was much faster than that of the negative control and *FpOAR*-transformed yeast which reflected its strong adaptability to aluminum stress. These results implied that *AtOT* might confer the aluminum tolerance to the recombinant yeast cell.

Some soluble proteins will chelate with metal ions under metal ion stress to reduce the toxic effects caused by metal ions. Therefore, the total protein (TP) content was measured as an indicator of the degree of toxicity of aluminum stress to cells (Figure 8D). The TP content of each transformant was similar under normal conditions. However, the TP content of the negative control significantly decreased and was lower than that of the *FpOAR-* and *AtOT-*transformed yeasts under 2.7 mM Al^3+^ stress, with it being 9014.5 μg/mL, and the TP content of the *FpOAR-* and *AtOT-*transformed yeasts decreased slightly, which were 10,327.6 μg/mL and 10,477.2 μg/mL, respectively. These results indicated that the adverse effect of Al^3+^ to the *FpOAR*- and *AtOT*- transformed yeasts was lesser than that of the negative control.

Malondialdehyde (MDA) is a product of membrane lipid peroxidation, and its content can be used to measure the resistance of yeast cells to various stresses. In general, the higher the intracellular MDA content, the higher the degree of plasma membrane damage. The MDA content of each transformant was similar when there was no aluminum stress. Under 2.7 mM Al^3+^ stress, the MDA content of the negative control was significantly higher than that of the *FpOAR*- and *AtOT*-transformed yeasts, thereby reaching 0.172 μM, which was 2.11 times higher than that under normal conditions. In contrast, the MDA content of the *FpOAR*- and *AtOT*-transformed yeasts increased slightly, thereby reaching 0.089 μM and 0.104 μM, respectively, which were 1.12 times and 1.26 times higher than that under normal conditions (Figure 8E). These results indicated that the oxidative damage of the plasma membrane in *FpOAR*- and *AtOT*-transformed yeasts was less than that in the negative control under aluminum stress.

Peroxidase (POD) is widely found in animals, plants, and microorganisms. It is a marker enzyme of peroxisomes that plays an important role in physiological processes such as scavenging reactive oxygen species (ROS), disease resistance, salt resistance, and cell death. No significant difference was observed in POD content among different transformants under normal conditions, as shown in Figure 8F. Under 2.7 mM Al^3+^ stress, the POD content of the negative control increased to 0.961 U/mg protein, which was significantly higher than that of the *FpOAR*- and *AtOT*-transformed yeasts and was 2.05 times of that under normal conditions. However, the POD content of *FpOAR*-transformed yeast increased to 0.71 U/mg protein, which was 1.43 times of that under normal conditions. The POD content of the *AtOT*-transformed yeast did not change significantly compared with that under normal conditions, which was 0.445 U/mg protein. These results suggest that the ROS content and oxidative damage degree in the negative control were higher than those of the *FpOAR*- and *AtOT*-transformed yeasts, which further proved that *AtOT* can detoxify aluminum.

## 3. Discussion

The mechanism of roots secreting organic acids to detoxify aluminum has been studied in various plants. Among them, malic *AtMATE* and citric acid transporter genes *AtALMT1* of the model plant *Arabidopsis* guide help in the identification of organic acid transporter genes in other plants. However, the oxalate transporter gene in *Arabidopsis* has not been identified and reported until now. The candidate oxalate transporter gene *AtOT* was obtained from *Arabidopsis* through BLASTP homologous alignment. We performed bioinformatics analysis and investigated the effects of *AtOT* on oxalic acid secretion and aluminum tolerance in yeast and *Arabidopsis* to verify the function of *AtOT*.

The prediction analysis of the conserved domain showed that the *AtOT* protein had SNARE-assoc conserved domains, belonging to the SNARE superfamily. Unlike MATE and ALMT proteins which are only involved in a few specific substrates such as citric acid and malic acid, the transport types and substrates of SNARE proteins are very complex [43,44]. At present, the research on plant SNARE proteins is still in its infancy, many scholars believe that the SNARE proteins primarily play a role in the transport of the inner membrane system by membrane fusion, including regulating vesicle synthesis, directional transport, and identifying and promoting the fusion between vesicles and specific target membranes which are highly conserved in animals, plants, and fungi [45]. According to our previous research, AtOT is quite different from other identified plasma-membrane-localized plant SNARE proteins. It was clustered into the SNARE-assoc subfamily with *Hevea brasiliensis* oxalate transporter HbOT1 and HbOT2 [42]. Combined with the results of subcellular localization and function characterization in yeast, we concluded that the *AtOT* protein may play a role in the membrane vesicle transport of certain substances such as oxalic acid. Moreover, the *AtOT* protein, located in the plasma membrane, has five transmembrane helices and has a larger hydrophobic part than the hydrophilic part, which is consistent with the basic characteristics of transporters and functional annotation of SNARE proteins.

The expression of *AtOT* in *Arabidopsis* wild-type Col-0 was significantly upregulated under aluminum stress, which increased first and then decreased with the increase in aluminum concentration and increased with the aluminum treatment time, suggesting that *AtOT* is inducible by active Al^3+^ ions. Meanwhile, the knockout of *AtOT* hurt the growth of the roots, and aluminum stress amplified the adverse effect. Therefore, *AtOT* may play an important role in root growth and is closely related to the mechanism of aluminum detoxification in *Arabidopsis*.

The oxalic acid resistance and aluminum tolerance of *AtOT* was identified by the yeast AD1-8 system. On the colony phenotype, the *AtOT*-transformed yeast showed stronger oxalic acid resistance and aluminum tolerance than the negative control under oxalic acid and aluminum stress. The lower pH value of the culture medium and higher dry weight of the yeast cells indicated that the *AtOT*-transformed yeast had better adaptability than the negative control in the liquid environment with 2 mM oxalic acid stress. The similar trend of the pH value and faster growth rate to the positive control FpOAR revealed the possibility of AtOT as an oxalate transporter. Based on the determination of oxalic acid content in vitro and vivo, we found that the AtOT and positive control FpOAR could help yeast cells overcome the continuous injury caused by oxalic acid stress and improve the oxalic acid metabolism rule of yeast cells under oxalic acid stress. The similar change in oxalic acid contents indicates that AtOT may play an important role in the transport and efflux of oxalic acid. The in vitro incubation of yeast membrane vesicles with [^13^C]oxalic acid verifying that AtOT regulates the absorption of oxalic acid by membrane vesicles, which further confirmed that AtOT is a SNARE family protein that could regulate the transport and efflux of oxalate by the membrane vesicle transport in *Arabidopsis*. Meanwhile, *AtOT*-transformed yeast entered the platform stage earlier under aluminum stress as compared with the negative control and *FpOAR*-transformed yeast, thereby reflecting its stronger adaptability to aluminum stress. All these results suggest that AtOT may act as an oxalic acid transporter involved in the *Arabidopsis* aluminum detoxification.

Total protein and POD contents are important indices to measure the degree of oxidative damage in cells, while MDA content is an important index to measure the degree of oxidative damage in the plasma membrane. Correspondingly, they are used to examine the oxidative damage of yeast cells by Al^3+^ in this study. Al^3+^ inevitably caused the oxidation and destruction of the plasma membrane by binding to membrane proteins, competing for receptor binding sites, and seizing ion channels. If AtOT primarily detoxifies aluminum from the inside of the cell by the compartmentation of vacuoles or the production of specific proteases [46], it will lead to an increase in MDA content. The total protein content, POD, and MDA in the *AtOT*-transformed yeast were more stable than those in the negative control before and after aluminum stress. Hence, we concluded that AtOT can significantly reduce aluminum toxicity in vivo and plasma membranes and may play an important role in the external exclusion mechanism of aluminum detoxification by regulating the efflux of oxalic acid.

## 4. Materials and Methods

### 4.1. Materials and Treatments

The homozygous *Arabidopsis atot* mutant (SALK_002559C), in which the *AtOT* gene is knocked out, was purchased from the AraShare *Arabidopsis* mutant library (http://www.arashare.cn/index/ (accessed on 10 November 2021)), and the *Arabidopsis* wild type Col-0 was used in this study as the control.

The seeds were sterilized in 75% ethanol for 2 min, 95% ethanol for 2 min, and 10% NaClO for 10 min and washed five times with ddH_2_O before culture. Then, the disinfected seeds were sown on 1/2 MS solid medium, vernalized at 4 °C for 48 h in the dark, and then cultured in a light incubator. The light incubator program was set to 22 °C, a relative humidity of 70%, and a photoperiod of 16 h/8 h (day/night). After growing two true leaves, the seedlings were transferred to 1/2 MS solid medium (pH = 4.2) containing Al^3+^. The wild-type Col-0 seedlings treated with 0 (CK), 25, 50, 100, 150, and 200 μM Al^3+^ for 48 h and treated with 100 μM Al^3+^ for 0 (CK), 3, 6, 12, 24, and 48 h were collected to extract RNA for expression analysis of *AtOT* in response to the concentration and stress time of aluminum, respectively.

The yeast mutant strain AD12345678 (AD1-8) was donated by Professor Tang of Shanghai JiaoTong University and Professor Richard Cannon of Otago University.

### 4.2. RNA Extraction, cDNA Synthesis, and qRT-PCR Analysis

Total RNA was extracted from the seedlings of *Arabidopsis* using a TIANGEN plant total RNA extraction kit (Tiangen Biochemical Technology Co., Ltd., Beijing, China) and cDNA was synthesized using TaKaRa reverse transcription kit (Baori Doctor Technology Co., Ltd., Dalian, China). The concentration and purity of RNA and cDNA were analyzed by a NanoDrop 2000 ultra-micro nucleic acid protein analyzer (Thermo Fisher Technology (China) Co., Ltd., Shanghai, China). The qRT-PCR analysis was performed on a CFX96 TOUCH real-time fluorescent quantitative PCR instrument (Bio-Rad, Hercules, CA, USA). The internal reference gene *AtActin2* (AT3G18780) was selected to normalize the data. The quantitative variations of gene expression were analyzed by the 2^−ΔΔCT^ method and the specific primers are listed in Appendix A.

### 4.3. Cloning and Bioinformatics Analysis

The ORF sequences of *AtOT* were amplified by PCR using specific primers (Appendix A) with the cDNA of *Arabidopsis* wild-type Col-0 as a template and the fragment was connected to the pMD-18T vector for sequencing.

The online website ProtParam (https://web.expasy.org/protparam/ (accessed on 5 May 2022)) was used to predict the basic physicochemical properties of the *AtOT* protein, including the molecular weight (MW), isoelectric point (pI), instability index (II), and total average hydrophilicity (GRAVY) of *AtOT*. The transmembrane region was predicted with TMPRED 2.0 (https://services.healthtech.dtu.dk/service.php?TMHMM-2.0 (accessed on 5 May 2022)). Amino acid hydrophilic/hydrophobic analysis was carried out based on ProtScale (http://web.expasy.org/cgi-bin/protscale/protscale.pl (accessed on 5 May 2022)). The subcellular location was predicted using PSORT (https://www.genscript.com/psort.html (accessed on 5 May 2022)). The conserved domains of *AtOT* protein were analyzed using the NCBI CDD database (https://www.ncbi.nlm.nih.gov/cdd (accessed on 7 May 2022)) and SMART (http://smart.embl-heidelberg.de/ (accessed on 8 May 2022)). All of the gene and protein sequences involved in this study were downloaded from the NCBI database (https://www.ncbi.nlm.nih.gov/ (accessed on 7 June 2020)).

### 4.4. Subcellular Localization of AtOT in N. benthamiana

In the subcellular localization assay, the cDNA of *AtOT* was subcloned into the expression vector pCAMBIA1300:GFP under the control of the 35S promoter using the specific primers listed in Appendix A. The pCAMBIA1300:AtOT-GFP fusion vector and pCAMBIA1300:GFP were separately introduced into the *Agrobacterium tumefaciens* strain GV3101 by the heat shock method. The *A. tumefaciens* suspensions were subsequently injected into leaves of 4-week-old *Nicotiana benthamiana* plants and the transient expression of *AtOT* was detected after 48 h using LSM800 a confocal laser scanning microscope (Carl Zeiss Shanghai Co. Ltd., Shanghai, China).

### 4.5. Homozygous Identification and Reverse Genetic Analysis of AtOT in Arabidopsis

To confirm the *atot* mutant homozygotes, total DNA was extracted by the CTAB method from the different *atot* mutant plants cultured for 3 weeks. Then, the three-primer method, i.e., the LP, RP, and LBb1.3 primers (Appendix A) designed by the Genomic Analysis Laboratory of the Salk Institute (http://signal.salk.edu/ (accessed on 7 January 2022)), was utilized for the identification of the *atot* mutant homozygous plants. The plants which have a clear band for the LBb1.3 + RP but have no band for the LP + RP primers during PCR were identified to be homozygous atot mutants and utilized in the subsequent experiments [47]. The Col-0 and *atot* mutant seedlings under 0 (CK), 25, 50, 100, 150, and 200 μM Al^3+^ stresses for 2 weeks were randomly selected for root length recording and the determination of proline (Pro) and malondialdehyde (MDA). Pro content was determined using the proline quantitative assay kit (Nanjing Jiancheng Bioengineering Institute Co., Ltd., Nanjing, China). MDA content was determined by TBA colorimetry [48].

### 4.6. Oxalic Acid Resistance and Aluminum Tolerance of AtOT in Yeast

The ORF sequences of *AtOT* were cloned into the pDR196 vector (Zhuang Meng International Biogene Technology Co., Ltd., Beijing, China) to generate the construct pDR196-*AtOT* by homologous recombination using specific primers (Appendix A) and then transformed into yeast mutant AD1-8 (Δ*yor1*, Δ*snq2*, Δ*pdr5*, Δ*pdr10*, Δ*pdr11*, Δ*ycf1*, Δ*pdr3*, and Δ*pdr15*) [49] by lithium acetate transformation. The yeast transformants were cultured in SD (-Ura) (Aili Biotechnology Co., Ltd., Shanghai, China) liquid medium at 30 °C with 180 rpm until OD_600_ = 0.5 and diluted in a 10-fold gradient with ddH_2_O. Then, they were grown on SD (-Ura) plates containing 0, 2, 4, 8, and 10 mM oxalic acid for 4 days, respectively. The colony phenotypes of the yeast cells at 2.4, 2.5, 2.6, 2.7, and 2.8 mM Al^3+^ were recorded by the same method.

Transformants cultured with OD_600_ = 1.0 were extracted and added in SD (-Ura) liquid medium containing 2 mM oxalic acid and cultured at 30 °C with 180 rpm for 13 days to further study the oxalic acid resistance of *AtOT*. Subsequently, the bacterial solution was centrifuged at 1000× *g* for 10 min to separate the yeast cells from the medium after being washed twice with ddH_2_O to determine the oxalic acid content in the yeast cells and the medium using boxbio oxalic acid (OA) content determination kit (Box Biotechnology Co., Ltd., Beijing, China). The pH of the bacterial solution was determined by a METTLER TOLEDO FE plus pH meter (Metler Toledo Technology Co., Ltd., Shanghai, China). A SCIENTZ-30YG/A freeze-drying machine (Xinzhi Freeze-drying Equipment Co., Ltd., Ningbo, China) was used for freeze-drying the yeast cells.

Similarly, each transformant was cultured in a 50 mL SD (-Ura) liquid medium containing 2.7 mM Al^3+^ for 48 h to further study the aluminum resistance mechanism of *AtOT*. The OD_600_ value of the bacterial solution was measured every 2 h to plot the growth curve. Then, the bacterial solution was centrifuged at 1000× *g* for 10 min, and yeast cells were collected to determine the total protein (TP), malondialdehyde (MDA), and peroxidase (POD). TP and POD content were determined using a protein quantitative assay kit (Nanjing Jiancheng Bioengineering Institute Co., Ltd., Nanjing, China) and a peroxidase assay kit (Keming Biotechnology Co., Ltd., Suzhou, China), respectively.

Subsequently, yeast membrane vesicles for the in vitro oxalate transport study were isolated with the following methods. Yeast transformants were cultured in 50 mL SD (-Ura) liquid medium until OD_600_ = 2.0. After being washed twice with ddH_2_O, the cells were resuspended in 50 mL spheroplast buffer (1.1 M sorbitol, 20 mM Tris-HCl [pH = 7.6], and 1 mM DTT containing 50 U of Lyticase 20 T/mL) for 10 min to digest the cell wall. Afterwards, spheroplasts were collected by 1200× *g* for 10 min, then resuspended and shocked gently in 25 mL breaking buffer (1.1 M glycerol, 50 mM Tris-ascorbate [pH = 7.4], 5 mM EDTA, 1 mM DTT, 1.5% polyvinylpyrrolidone, 2 mg/mL BSA, 1 mM PMSF, 10 μg/mL leupeptin, 2 μg/mL aprotinin, and 2 μg/mL pepstatin) on ice. The solution was centrifuged twice at 3000× *g* for 10 min to remove unbroken cells and other impurities; then, the supernatant was centrifuged at 100,000× *g* for 1 h to obtain yeast membrane vesicles which were suspended subsequently to a protein concentration of 5 mg/mL in vesicle buffer (1.1 M glycerol, 50 mM Tris-MES [pH = 7.4], 1 mM EDTA, 1 mM DTT, 2 mg/mL BSA, 1 mM PMSF, 1 μg/mL leupeptin, 2 μg/mL aprotinin, and 2 μg/mL pepstatin).

Furthermore, vesicles (100 μg of protein) were incubated in 125 μL transport buffer (0.4 M glycerol, 100 mM KCl, 20 mM Tris-MES [pH = 7.4], 1 mM DTT, 0 (CK), or 5 mM MgATP) containing 0.2 mM [^13^C]oxalic acid at 25 °C for 10 min, and the incubation solution was injected into syringe filters (33 mm in diameter and 0.45-μm pore size; Labgic Technology Co., Ltd., Beijing, China) and washed thrice with 1 mL transport buffer by needle-free injectors to absorb yeast vesicles. For the measurement of δ ^13^C, 0.001 g silicon algae were added to 10 μL filtrate and then freeze-dried until the water was completely removed and transferred to a tin cup (6 mm × 4 mm) by a Flash EA 1112 Automatic Element Analyzer (Thermo Fisher Scientific Co., Ltd., Shanghai, China).

### 4.7. Statistical Methods

The gene expression level was from three biologicals with three technical repetitions. In addition, the pH value, dry weight, oxalic acid content, total protein content, malondialdehyde content, and peroxidase content of the yeast were the mean ± standard deviation of three biological replicates. Moreover, Microsoft Office Excel was used for data analysis and mapping. IBM SPSS Statistics 25 was used for a single-factor ANOVA test to analyze the difference. Duncan’s method was used for the analysis.

## 5. Conclusions

In this study, the candidate oxalate transporter gene *AtOT* was cloned and identified from *Arabidopsis*. This gene can make yeast cells obtain stronger oxalate resistance, and we believe that this phenomenon is caused by its induction of oxalic acid efflux. On the other hand, it can promote the yeast aluminum tolerance to higher concentrations and can be induced by aluminum stress in *Arabidopsis*. It can regulate the membrane vesicle transport of oxalate to avoid the oxidative damage of Al^3+^. The knockout of *AtOT* will inhibit the root growth of *Arabidopsis* under normal conditions, and this inhibition will be further amplified under aluminum stress. All of these results suggest that *AtOT* likely played a pivotal role in regulating oxalate-related vesicle transport and the mechanism of oxalate efflux to detoxify aluminum; more solid research is worthwhile for the *AtOT* gene.

## Figures and Tables

**Figure 1 ijms-24-04516-f001:**
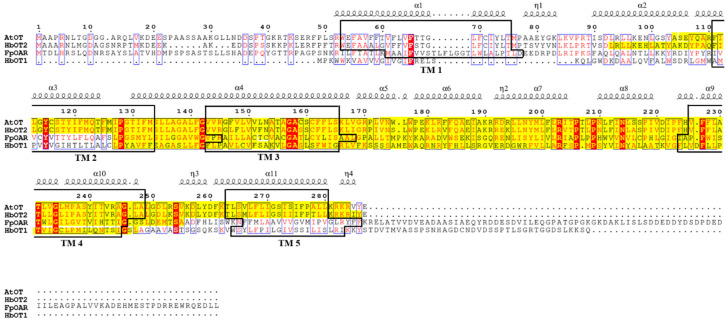
Multiple sequence alignment of FpOAR, HbOT1, HbOT2, and *AtOT*. Different secondary structures are labelled above a specific sequence, black boxes represent the transmembrane domains (TM1–TM5) of the protein, and the yellow highlighted fragment represents the SNARE-assoc conserved domain of the protein. Conserved and similar residues are identified by red shadows and blue boxes, respectively.

**Figure 2 ijms-24-04516-f002:**
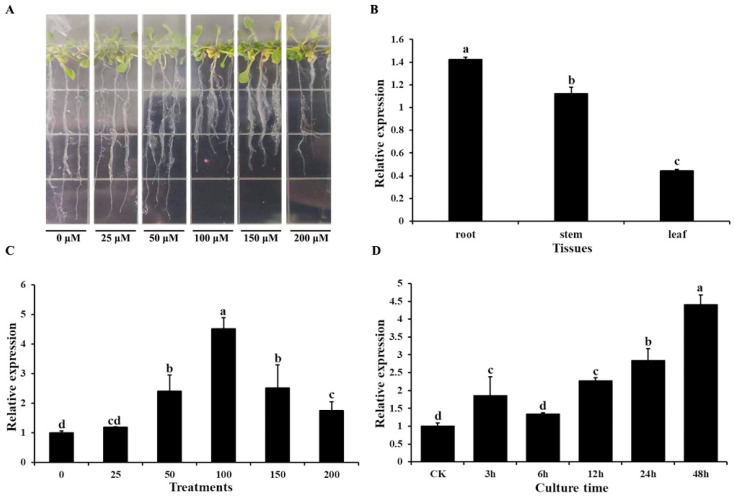
Expression pattern of the *AtOT* gene in *Arabidopsis*. (**A**) Phenotype of wild-type Col-0 under 0 (CK), 25, 50, 100, 150, and 200 μM of AlCl_3_·6H_2_O for 48 h. (**B**) qRT-PCR analysis of *AtOT* transcripts in different tissues. (**C**) qRT-PCR analysis of *AtOT* under 0 (CK), 25, 50, 100, 150, and 200 μM Al^3+^ for 48 h in the root of *Arabidopsis* wild-type Col-0. (**D**) qRT-PCR analysis of *AtOT* under 100 μM Al^3+^ for 0 (CK), 3, 6, 12, 24, and 48 h in the root of *Arabidopsis* wild-type Col-0. Different letters above the bars indicate significant differences among the treatments at *p* < 0.05.

**Figure 3 ijms-24-04516-f003:**
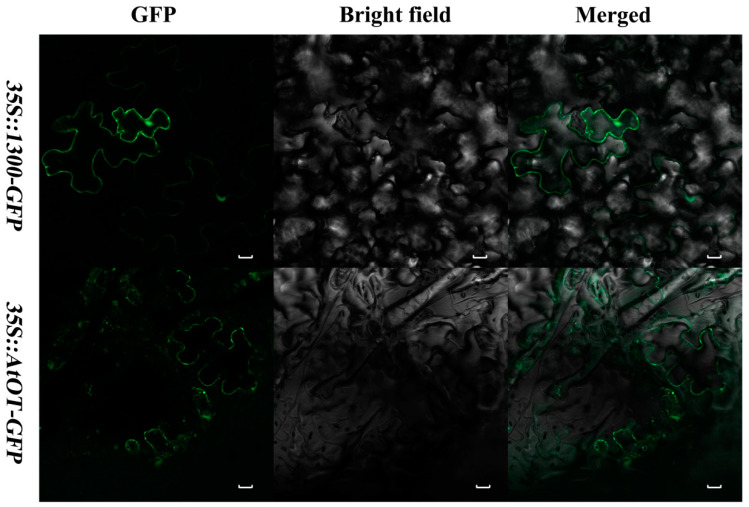
Subcellular localization of *AtOT* in tobacco leaves. The *35S::AtOT-GFP* fusion protein was transiently expressed in tobacco (*N. benthamiana*) leaf epidermal cells. The bars indicate the length of 20 um.

**Figure 4 ijms-24-04516-f004:**
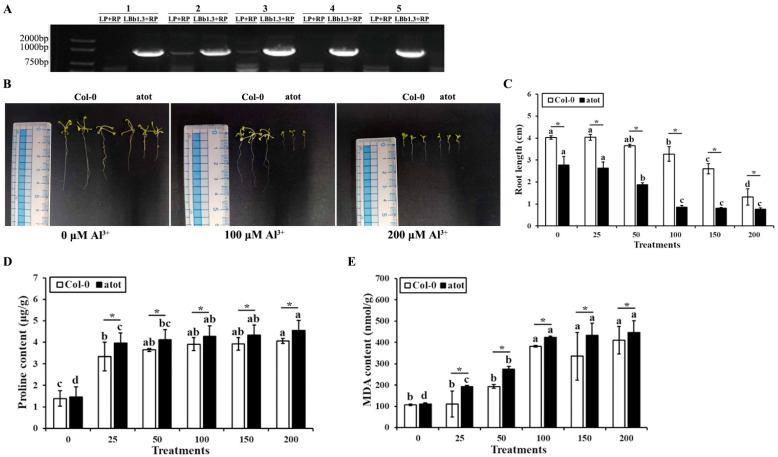
Effects of aluminum on the wild-type and *atot* mutant *Arabidopsis*. (**A**) Confirmation of the homozygosity of the *atot* mutant plants. Information on the *atot* mutant (SALK_002559C) can be found at http://signal.salk.edu/cgi-bin/tdnaexpress/ (accessed on 7 January 2022), and plants no. 1, 4, and 5 which showed a clear band for the LBb1.3 + RP primers but no bands for the LP + RP primers in the three primers PCR method were identified as homozygotes and utilized for the subsequent experiments. (**B**) Phenotype of the wild-type Col-0 and *atot* mutant under 0 (CK), 100, and 200 μM Al^3+^ for 2 weeks. (**C**) Root length comparison of the wild-type Col-0 and *atot* mutant under 0 (CK), 25, 50, 100, 150, and 200 μM Al^3+^ for 2 weeks. (**D**) Proline content of the wild-type Col-0 and *atot* mutant under 0 (CK), 25, 50, 100, 150, and 200 μM Al^3+^. (**E**) Malonaldehyde content of the wild-type Col-0 and *atot* mutant under 0 (CK), 25, 50, 100, 150, and 200 μM Al^3+^. Different letters above the bars indicate significant differences among the treatment concentrations and the asterisk (*) represents significant differences between the wild-type Col-0 and *atot* mutant *Arabidopsis* at *p* < 0.05.

**Figure 5 ijms-24-04516-f005:**
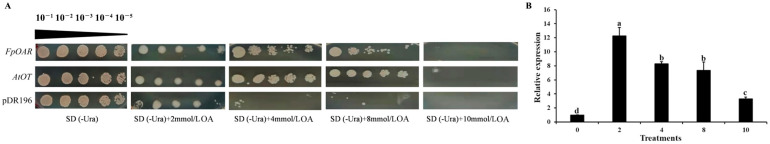
Growth of yeast transformants under different concentrations of oxalic acid. (**A**) Colony phenotype of yeast transformants under different concentrations of oxalic acid for 4 days. The yeast concentrations for each treatment were 10^−1^, 10^−2^, 10^−3^, 10^−4^, and 10^−5^ from left to right. (**B**) qRT-PCR analysis of *AtOT* under 0 (CK), 2, 4, 8, and 10 mM oxalic acid stresses in yeast cells. Different letters above the bars indicate significant differences among treatments at *p* < 0.05.

**Figure 6 ijms-24-04516-f006:**
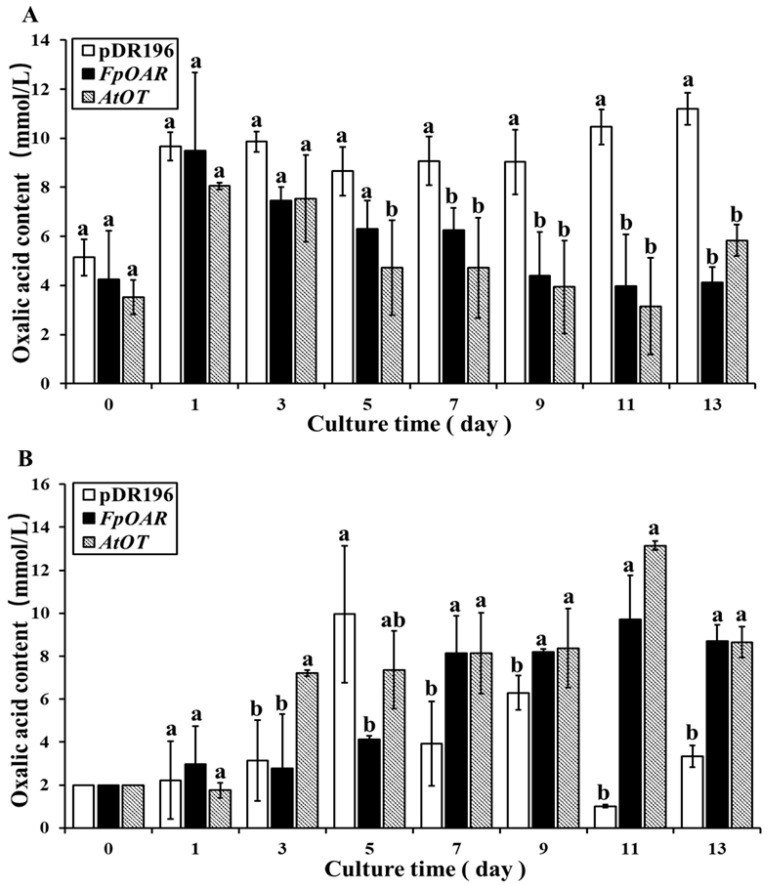
Changes in oxalic acid contents in the medium and cells of the yeast transformants under 2 mmol/L oxalic acid stress. (**A**) Changes in oxalic acid contents in the yeast transformants under 2 mM oxalic acid stress. (**B**) Changes in oxalic acid contents in the culture medium under 2 mM oxalic acid stress. Different letters indicate significant differences among different recombinant yeast cells at *p* < 0.05.

**Figure 7 ijms-24-04516-f007:**
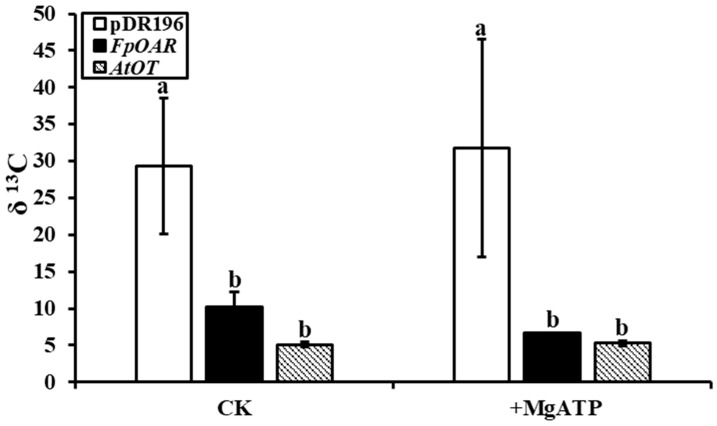
[^13^C]oxalic acid residue in the filtrate of pDR196- (negative control), *FpOAR*- (positive control), and *AtOT*-transformed yeast membrane vesicles. Different letters in the figure indicated significant differences among the recombinant yeast cells at *p* < 0.05.

**Figure 8 ijms-24-04516-f008:**
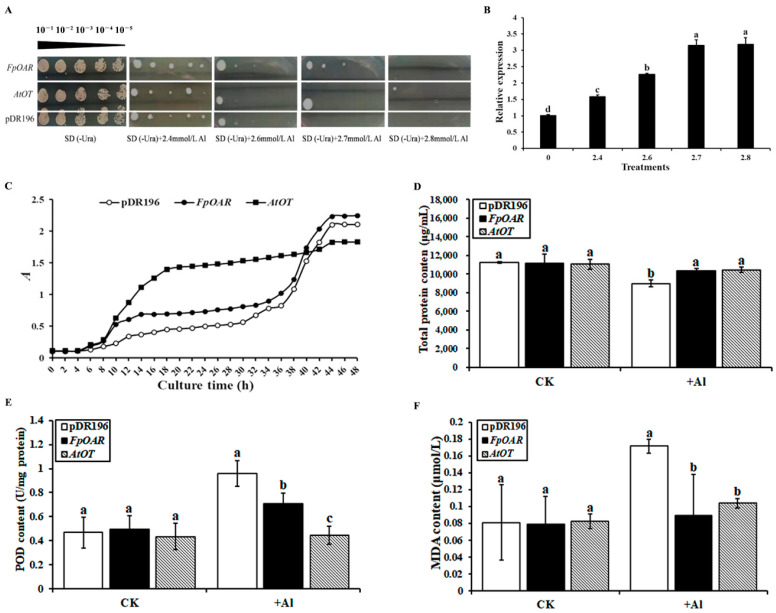
Effects of *AtOT* on yeast cell aluminum tolerance. (**A**) Effect of *AtOT* aluminum on yeast cell colony phenotype. (**B**) qRT-PCR analysis of *AtOT* in the *AtOT*-recombinant yeast cells under 0 (CK), 2.4, 2.6, 2.7, and 2.8 mM Al^3+^. (**C**) Growth curve of yeast transformants under 2.7 mM Al^3+^ stress. (**D**) Determination of total protein content of yeast transformants under 2.7 mM Al^3+^ stress for 48 h. (**E**) Determination of malondialdehyde (MDA) content in yeast transformants under 2.7 mM Al^3+^ stress for 48 h. (**F**) Determination of peroxidase (POD) content in yeast transformants under 2.7 mM Al^3+^ stress for 48 h. Different letters above bars in (**C**–**E**) indicate significant differences among various recombinant yeast cells at *p* < 0.05.

**Table 1 ijms-24-04516-t001:** Variation of culture solution pHs for different recombinant yeast cells under 2 mM oxalic acid stress.

Yeast Cells	Culture Time (Day)							
	0	1	3	5	7	9	11	13
pDR196	5	4.59 ± 0.082 Aa	3.87 ± 0.095 Ba	3.44 ± 0.125 Ba	3.1 ± 0.066 Ca	2.89 ± 0.118 Da	2.78 ± 0.046 Da	2.76 ± 0.072 Da
*FpOAR*	5	4.09 ± 0.075 Ab	3.19 ± 0.075 Bab	2.76 ± 0.095 Cb	2.64 ± 0.03 Db	2.55 ± 0.053 DEb	2.5 ± 0.066 EFb	2.42 ± 0.05 Fb
*AtOT*	5	4.15 ± 0.089 Ab	3.16 ± 0.066 Bb	2.7 ± 0.066 Cb	2.55 ± 0.108 CDb	2.46 ± 0.044 Db	2.43 ± 0.066 Db	2.38 ± 0.053 Db

Note: Different uppercase and lowercase letters show significant differences among the treatment times and recombinant yeast cells, respectively (*p* < 0.05).

**Table 2 ijms-24-04516-t002:** Variation of dry weight for different recombinant yeast cells under 2 mM oxalic acid stress. (g).

Yeast Cells	Culture Time (Day)						
	1	3	5	7	9	11	13
pDR196	0.025 ± 0.01 Ac	0.05 ± 0.01 Ab	0.125 ± 0.018 Bc	0.175 ± 0.015 Cb	0.175 ± 0.02 Cc	0.2 ± 0.023 Cb	0.2 ± 0.015 Cc
*FpOAR*	0.05 ± 0.015 Ab	0.125 ± 0.02 Ba	0.175 ± 0.015 Cb	0.225 ± 0.018 Da	0.225 ± 0.018 Db	0.225 ± 0.018 Db	0.25 ± 0.01 Db
*AtOT*	0.125 ± 0.023 Aa	0.15 ± 0.026 Aa	0.225 ± 0.017 Ba	0.25 ± 0.018 BCa	0.275 ± 0.025 CDa	0.3 ± 0.026 Da	0.35 ± 0.027 Ea

Note: Different uppercase and lowercase letters indicate significant differences among the treatment times and recombinant yeast cells, respectively (*p* < 0.05).

## Data Availability

All data are available on reasonable request to the corresponding authors.

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
