# Peer review of "An Oxalate Transporter Gene, AtOT, Enhances Aluminum Tolerance in Arabidopsis thaliana by Regulating Oxalate Efflux"

_ijms, 2023, doi:10.3390/ijms24054516_

Round 1

Reviewer 1 Report

This manuscript aimed to study AtOT up-regulated in response to aluminum stress and enhancing aluminum tolerance in Arabidopsis thaliana by regulating oxalate efflux. Phenotype identifications of AtOT enhancing oxalic acid resistance and aluminum tolerance were carried out in yeast cells and Arabidopsis, respectively. The manuscript is well written, and for most part the data is well presented and interpreted. However, there are a few issues required to be addressed to be fully justified in the manuscript.

1. Analysis of AtOT expression pattern in Figure 2B should be carried out by qRT-PCR analysis under 0 (CK), 25, 50, 100, 150, 200 μM Al3+ stresses.

2. The determination results of proline and MDA contents should be added in Figure 6 “Aluminum tolerance identification of AtOT in Arabidopsis”.

3.  Several units need to be corrected thorough the text and the markers in Figure3, 5 and 6. Such as: “umol/L” to “μM”, “mmol/L” to “mM”, “uM” to “μM”, “ug” to “μg”.

4.  The manuscript should be modified by an English native specialist in grammars to make it more readable.

Author Response

Dear reviewer1,

Thank you for your valuable comments on our manuscript 2071419 “A candidate oxalate transporter gene, AtOT, enhances aluminum tolerance in Arabidopsis thaliana by regulating oxalate efflux”. We have carefully revised the paper according to you and other reviewers’ comments. Those comments helped us a lot to improve our manuscript. A point by point response to reviewers is attached at the end of this letter.  

We think the paper can be now published in the International Journal of Molecular Sciences. We are looking forward for your further evaluation.

Kind regards

Feng

This manuscript aimed to study AtOT up-regulated in response to aluminum stress and enhancing aluminum tolerance in Arabidopsis thaliana by regulating oxalate efflux. Phenotype identifications of AtOT enhancing oxalic acid resistance and aluminum tolerance were carried out in yeast cells and Arabidopsis, respectively. The manuscript is well written, and for most part the data is well presented and interpreted. However, there are a few issues required to be addressed to be fully justified in the manuscript.

  1. Analysis of AtOTexpression pattern in Figure 2B should be carried out by qRT-PCR analysis under 0 (CK), 25, 50, 100, 150, 200 μM Al3+ stresses.

Thanks for your suggestion. We have added the qRT-PCR analysis under 0 (CK), 25, 50, 100, 150, 200 μM Al3+ stresses in the manuscript.

  1. The determination results of proline and MDA contents should be added in Figure 6 “Aluminum tolerance identification of AtOT in Arabidopsis”.

Thanks for your reminder. We have added that in the manuscript.

  1. Several units need to be corrected thorough the text and the markers in Figure3, 5 and 6. Such as: “umol/L” to “μM”, “mmol/L” to “mM”, “uM” to “μM”, “ug” to “μg”.

Thanks for your reminder. We have modified that in the manuscript.

  1. The manuscript should be modified by an English native specialist in grammars to make it more readable.

Thanks for your suggestion. We have commissioned a professional organization to polish the language expression.

Reviewer 2 Report

This paper by Yang et al. reports a candidate transporter gene, AtOT, involved in oxalic acid resistance and aluminum (Al) tolerance in Arabidopsis. The authors found that the expression of AtOT was up-regulated by Al exposure, and AtOT showed efflux transport activity for oxalic acid in yeast. Finally, they exploited a T-DNA insertion mutant and identified that inhibition of AtOT resulted in a harmful effect on root elongation, which was further amplified by Al stress. Based on these results, the authors concluded that AtOT played an important role in the external exclusion mechanism of Al detoxification by regulating the efflux of oxalic acid. However, my main concern is that most data were not sufficient to support the conclusions as the author claimed. Here are the main points:

1.     In Fig.2, the enhanced expression of AtOT under Al stress was provided by qPCR, it is necessary to show whether its expression pattern and tissue localization are the same in Arabidopsis with Al stress. More evidences are required, such as histochemical GUS staining from transgenic lines of AtOT::GUS under the control of its own promotor. And in Fig. 2B, more concentrations of Al should be applied to confirm the AtOT expression.

2.     According to the author Graphical Abstract, AtOT mediated transport of oxalate in the vesicle, and may through vesicles transporters transport to the apoplast, therefore, the subcellular localization of AtOT in Arabidopsis should be supplemented. Additionally, the authors only showed data in yeast, I think other methods such as electrophysiological experiments should also be applied for avoiding the false positives. As an efflux transporter of oxalic acid, how AtOT affects Al accumulation and oxalic acid secretion in Arabidopsis root cells?

3.     In Fig. 3, the expression level of AtOT in yeast should be supplemented to verify the expression. In Fig.6, the authors used atot mutant to investigate Al sensitivity. How about Al accumulation and oxalic acid secretion in atot mutant with or without Al stress? As only one mutant was used here, complementary lines of AtOT in atot background and AtOT overexpressed Arabidopsis materials need to be provided. Moreover, when compared to WT plants, root elongation of atot mutant was repressed under normal conditions, how to explain this?

4.     There are some mistakes in English writing over the manuscript. For example, "umol/L" should be "µmol/L" (line 115). Please check English writing over the manuscript. In Fig. 3, the significant difference among treatments was evidenced by different letters but not *. The writing of the MS needs to be improved.

Author Response

Dear reviewer2,

Thank you for your valuable comments on our manuscript 2071419 “A candidate oxalate transporter gene, AtOT, enhances aluminum tolerance in Arabidopsis thaliana by regulating oxalate efflux”. We have carefully revised the paper according to you and other reviewers’ comments. Those comments helped us a lot to improve our manuscript. A point by point response to reviewers is attached at the end of this letter.  

We think the paper can be now published in the International Journal of Molecular Sciences. We are looking forward for your further evaluation.

Kind regards

Feng

This paper by Yang et al. reports a candidate transporter gene, AtOT, involved in oxalic acid resistance and aluminum (Al) tolerance in Arabidopsis. The authors found that the expression of AtOT was up-regulated by Al exposure, and AtOT showed efflux transport activity for oxalic acid in yeast. Finally, they exploited a T-DNA insertion mutant and identified that inhibition of AtOT resulted in a harmful effect on root elongation, which was further amplified by Al stress. Based on these results, the authors concluded that AtOT played an important role in the external exclusion mechanism of Al detoxification by regulating the efflux of oxalic acid. However, my main concern is that most data were not sufficient to support the conclusions as the author claimed. Here are the main points:

  1. In Fig.2, the enhanced expression of AtOT under Al stress was provided by qPCR, it is necessary to show whether its expression pattern and tissue localization are the same in Arabidopsis with Al stress. More evidences are required, such as histochemical GUS staining from transgenic lines of AtOT::GUS under the control of its own promotor. And in Fig. 2B, more concentrations of Al should be applied to confirm the AtOT expression.

Thanks for your suggestion. We have added the qRT-PCR analysis of AtOT  in root, stem, and leaf of Arabidopsis. In addition, we have added more aluminum treatment concentrations to confirm the expression pattern of AtOT.

  1. According to the author Graphical Abstract, AtOT mediated transport of oxalate in the vesicle, and may through vesicles transporters transport to the apoplast, therefore, the subcellular localization of AtOT in Arabidopsis should be supplemented. Additionally, the authors only showed data in yeast, I think other methods such as electrophysiological experiments should also be applied for avoiding the false positives. As an efflux transporter of oxalic acid, how AtOT affects Al accumulation and oxalic acid secretion in Arabidopsis root cells?

Thanks for the reminder. We have confirmed the subcellular localization of AtOT in tobacco (N. benthamiana) leaf epidermal cells and found that AtOT is a membrane localized protein. As you said, the electrophysiological experiments will be our next research focus, which will verify the transport channel activity of AtOT by two-electrode voltage clamp in Xenopus oocytes. However, we cannot include the results in this paper At present, our results basically support that AtOT gene is closely related to the efflux of oxalic acid. Based on the function of AtOT homologies SNARE _ asso proteins, we speculate that AtOT is most likely affects oxalic acid secretion by regulating vesicle transport in yeast cells. However, its mechanism of regulating oxalate efflux in Arabidopsis sitll requires furthe research.

  1. In Fig. 3, the expression level of AtOT in yeast should be supplemented to verify the expression.In Fig.6, the authors used atot mutant to investigate Al sensitivity. How about Al accumulation and oxalic acid secretion in atot mutant with or without Al stress? As only one mutant was used here, complementary lines of AtOT in atot background and AtOT overexpressed Arabidopsis materials need to be provided. Moreover, when compared to WT plants, root elongation of atot mutant was repressed under normal conditions, how to explain this?

Thanks for your reminder. We have supplemented the qRT-PCR analysis of AtOT in yeast under different concentration of oxalic acid and the determination of oxalic acid content in Arabidopsis under different concentration of Al stress. As we said, we treated Arabidopsis with aluminum stress at the concentration of uM, which makes the aluminum content in Arabidopsis too low for many color reaction-based aluminum measurement methods to accurately detect aluminum concentration. In the follow-up study, we will try to use ICP-AES to determine the aluminum content in Arabidopsis. Since the T3 generation of AtOT overexpresed Arabidopsis has not yet been screened, we are sorry that this part cannot be displayed in the manuscript. By repeating this experiment, we determined the difference between WT plants and atot mutant under normal conditions. We believe that the AtOT gene can also regulate the root growth of Arabidopsis through some biological pathways.

  1. There are some mistakes in English writing over the manuscript. For example, "umol/L" should be "µmol/L" (line 115). Please check English writing over the manuscript. In Fig. 3, the significant difference among treatments was evidenced by different letters but not *. The writing of the MS needs to be improved.

Thanks for your suggestion. We have modified that in the manuscript.

Reviewer 3 Report

The manuscript describes the AtOT as an oxalate transporter involved on aluminum tolerance in Arabidopsis. However, several points have to be deeply revised in the whole manuscript: from the title to the conclusion. Some comments are repeated along the text due to the occurrence in different parts of the manuscript, and also in the Comments to authors.  

The manuscript has to be carefully revised by a native English writing specialist. The phrase constructions are poor and a professional reviewer would improve a lot the manuscript.

The title has to be revised: AtOT is not a candidate gene once the gene was validated in this manuscript. If AtOT is an oxalate transporter its function is to regulate the oxalate efflux. Finally, the title should be objective and express the most important result of the manuscript.

The oxalic acid resistance should be revised in the whole manuscript. In fact, this terminology was used mainly in the experiment with yeast, whereas the candidate gene was validated in plant. Oxalic acid is a real stress that triggers important resistance mechanisms in plants? If so, this stress has to be described in the introduction. All excess of organic or inorganic molecules would cause a metabolic disorder in biological organisms, but it is important to differ an abiotic stress that inhibit plant development, such as Al toxicity, than an excess of oxalic acid used to evaluate yeast performance. Be careful in the whole manuscript, from the title to the conclusion.

All figure captions are incomplete. They need to explain what is in each panel: biological sample, a brief methodology, not only the “Identification of homozygous mutant”. What is in the panel a gel with an amplified DNA fragment? Of Arabidopsis? Yeast? Which line or strain… Revise all figures and tables. The figures/tables should be placed closed to their descriptions in the manuscript. And also, in order of which is cited.

Reorganize the results presentation in biological models: after AtOT cloning, present first the results of both lines of Arabidopsis (Col-0 and mutant) for gene expression and root development. Then present the results of yeast.

Expression pattern: clarify what is the biological model. First define the best Al concentration, then perform a time course experiment. The expression pattern of AtOT in the mutant is very important to de presented, not only the control line.

The context of many phrases is incomplete or vague, in many points only one word is missing to make sense. Please see detailed observations along the text.

Mainly in the item 2.3, the interpretation of the statistical significance is neglected, giving place for a slight difference, most of the time without an important effect in the overall conclusion. Additionally, discussion is mixed with the results in this section.

Lines 220-224: “These results indicated that the accumulation of aluminum ions in FpOAR-and AtOT-transformed yeast…”  This citation was based on total protein content, which is completely different from the Al content. In addition, there are methods to quantify specifically Al in biological tissues. So, the data does not support this statement, which is very complicated for the manuscript approval.

The statistical analysis should be carefully revised. As shown for panel D and E of Figure 5, the statistical analysis was performed separately for each condition, which is not adequate for comparisons that were made between these conditions. Several observations were made in a proper part of the text, but at least for these topics, the statistical analysis should be revised.

Finally, the scheme proposed in Fig 6D is not completely supported by your data.

Parts of the discussion was placed in the results. Organize a discussion combining all results, different of how it is organized, where each individual result is followed by a phrase of discussion.

Author Response

Dear reviewer3,

Thank you for your valuable comments on our manuscript 2071419 “A candidate oxalate transporter gene, AtOT, enhances aluminum tolerance in Arabidopsis thaliana by regulating oxalate efflux”. We have carefully revised the paper according to you and other reviewers’ comments. Those comments helped us a lot to improve our manuscript. A point by point response to reviewers is attached at the end of this letter.  

We think the paper can be now published in the International Journal of Molecular Sciences.. We are looking forward for your further evaluation.

Kind regards

Feng

The manuscript describes the AtOT as an oxalate transporter involved on aluminum tolerance in Arabidopsis. However, several points have to be deeply revised in the whole manuscript: from the title to the conclusion. Some comments are repeated along the text due to the occurrence in different parts of the manuscript, and also in the Comments to authors.

The manuscript has to be carefully revised by a native English writing specialist. The phrase constructions are poor and a professional reviewer would improve a lot the manuscript.

Thanks for your reminder. We have commissioned a professional organization to polish language expression.

The title has to be revised: AtOT is not a candidate gene once the gene was validated in this manuscript. If AtOT is an oxalate transporter its function is to regulate the oxalate efflux. Finally, the title should be objective and express the most important result of the manuscript.

Thanks for your suggestion. We have modified that in the manuscript.

The oxalic acid resistance should be revised in the whole manuscript. In fact, this terminology was used mainly in the experiment with yeast, whereas the candidate gene was validated in plant. Oxalic acid is a real stress that triggers important resistance mechanisms in plants? If so, this stress has to be described in the introduction. All excess of organic or inorganic molecules would cause a metabolic disorder in biological organisms, but it is important to differ an abiotic stress that inhibit plant development, such as Al toxicity, than an excess of oxalic acid used to evaluate yeast performance. Be careful in the whole manuscript, from the title to the conclusion.

Thanks. Oxalic acid resistance is mainly used in the yeast experiments as the fungal oxalic acid transporters identification. We think it is a fundamental process to functionalized as an oxalic acid transport. We have think over and revised the exprssion in some places but in some places it was not changed. In addtion, we have added in the introduction that oxalic acid was reported as an abiotic stress in Arabidopsis, such as in the study of oxalate decarboxylase (Oxdc) gene and oxalate oxidase (Oxo) gene. Further supplementary specifications have been added in the manuscript.

All figure captions are incomplete. They need to explain what is in each panel: biological sample, a brief methodology, not only the “Identification of homozygous mutant”. What is in the panel a gel with an amplified DNA fragment? Of Arabidopsis? Yeast? Which line or strain… Revise all figures and tables. The figures/tables should be placed closed to their descriptions in the manuscript. And also, in order of which is cited.

Thanks for your suggestion. We have modified that in the manuscript.

Reorganize the results presentation in biological models: after AtOT cloning, present first the results of both lines of Arabidopsis (Col-0 and mutant) for gene expression and root development. Then present the results of yeast.

Thanks for your suggestion. We have modified that in the manuscript.

Expression pattern: clarify what is the biological model. First define the best Al concentration, then perform a time course experiment. The expression pattern of AtOT in the mutant is very important to de presented, not only the control line.

Thanks for your suggestion. We have modified that in the manuscript.

The context of many phrases is incomplete or vague, in many points only one word is missing to make sense. Please see detailed observations along the text.

Thanks for your suggestion. We have carefully revised the mansuscirpt and  commissioned a professional organization to polish the language expression.

Mainly in the item 2.3, the interpretation of the statistical significance is neglected, giving place for a slight difference, most of the time without an important effect in the overall conclusion. Additionally, discussion is mixed with the results in this section.

Thanks for your suggestion. We have modified that in the manuscript.

Lines 220-224: “These results indicated that the accumulation of aluminum ions in FpOAR-and AtOT-transformed yeast…” This citation was based on total protein content, which is completely different from the Al content. In addition, there are methods to quantify specifically Al in biological tissues. So, the data does not support this statement, which is very complicated for the manuscript approval.

Thanks for your suggestion. We have tried to determine the Al concentration in yeast, however,  the limit of dection for Al concentraton could not be reached by the color reaction mothod. We will try to use ICP-AES method to determine the aluminum content in yeast cells in our following research, but it is a pity that we cannot include the result in this manuscript.

The statistical analysis should be carefully revised. As shown for panel D and E of Figure 5, the statistical analysis was performed separately for each condition, which is not adequate for comparisons that were made between these conditions. Several observations were made in a proper part of the text, but at least for these topics, the statistical analysis should be revised.

Thanks for your suggestion. We have modified that in the manuscript.

Finally, the scheme proposed in Fig 6D is not completely supported by your data.

Thanks for your suggestion. In fact, it is only a mechanism to deduce the transmembrane transport of oxalic acid by AtOT gene, which we think will help readers to understand the Al detoxification mechanism.

Parts of the discussion was placed in the results. Organize a discussion combining all results, different of how it is organized, where each individual result is followed by a phrase of discussion.

Thanks for your suggestion. We have modified that in the manuscript.

Round 2

Reviewer 1 Report

Accept.

Author Response

Dear Reviewer 1,

Thanks for the positive feedback.

Kind regards,

Feng